# Direct economic burden of mental health disorders associated with polycystic ovary syndrome: Systematic review and meta-analysis

Surabhi Yadav[1], Olivia Delau[1], Adam J Bonner[2], Daniela Markovic[3], William Patterson[4], Sasha Ottey[4], Richard P Buyalos[5], Ricardo Azziz[2,6,7,8]*

[1]School of Global Public Health, New York University, New York, United States; [2]Department of Obstetrics and Gynecology, Heersink School of Medicine, University of Alabama at Birmingham, Birmingham, United States; [3]Division of General Internal Medicine and Health Services Research, University of California, Los Angeles, Los Angeles, United States; [4]PCOS Challenge: The National Polycystic Ovary Syndrome, Atlanta, United States; [5]Department of Obstetrics and Gynecology, University of California, Los Angeles, Los Angeles, United States; [6]Department of Health Policy, Management and Behavior, University at Albany, State University of New York, Rensselaer, United States; [7]Department of Medicine, Heersink School of Medicine, University of Alabama at Birmingham, Birmingham, United States; [8]Department of Healthcare Organization and Policy, School of Public Health, University of Alabama at Birmingham, Birmingham, United States

*For correspondence:
razziz67@gmail.com

## Abstract

## Abstract

**Background:** Polycystic ovary syndrome (PCOS) is the most common hormone disorder affecting about one in seven reproductive-aged women worldwide and approximately 6 million women in the United States (U.S.). PCOS can be a significant burden to those affected and is associated with an increased prevalence of mental health (MH) disorders such as depression, anxiety, eating disorders, and postpartum depression. We undertook this study to determine the excess economic burden associated with MH disorders in women with PCOS in order to allow for a more accurate prioritization of the disorder as a public health priority.

**Methods:** Following PRISMA reporting guidelines for systematic review, we searched PubMed, Web of Science, EBSCO, Medline, Scopus, and PsycINFO through July 16, 2021, for studies on MH disorders in PCOS. Excluded were studies not in humans, without controls, without original data, or not peer reviewed. As anxiety, depression, eating disorders, and postpartum depression were by far the most common MH disorders assessed by the studies, we performed our meta-analysis on these disorders. Meta-analyses were performed using the DerSimonian–Laird random effects model to compute pooled estimates of prevalence ratios (PRs) for the associations between PCOS and these MH disorders and then calculated the excess direct costs related to these disorders in U.S. dollars (USD) for women suffering from PCOS in the U.S. alone. The quality of selected studies was assessed using the Newcastle-Ottawa Scale.

**Results:** We screened 78 articles by title/abstract, assessed 43 articles in full text, and included 25 articles. Pooled PRs were 1.42 (95% confidence interval [CI]: 1.32–1.52) for anxiety, 1.65 (95% CI: 1.44–1.89) for depression, 1.48 (95% CI: PR: 1.06–2.05) for eating disorders, and 1.20 (95% CI:

0.96–1.50) for postpartum depression, for PCOS relative to controls. In the U.S., the additional direct healthcare costs associated with anxiety, depression, and eating disorders in PCOS were estimated to be $1.939 billion/yr, $1.678 billion/yr, and $0.644 billion/yr in 2021 USD, respectively. Postpartum depression was excluded from the cost analyses due to the non-significant meta-analysis result. Taken together, the additional direct healthcare costs associated with anxiety, depression, and eating disorders in PCOS were estimated to be $4.261 billion/yr in 2021 USD.

**Conclusions:** Overall, the direct healthcare annual costs for the most common MH disorders in PCOS, namely anxiety, depression, and eating disorders, exceeds $4 billion in 2021 USD for the U.S. population alone. Taken together with our prior work, these data suggest that the healthcare-related economic burden of PCOS exceeds $15 billion yearly, considering the costs of PCOS diagnosis, and costs related to PCOS-associated MH, reproductive, vascular, and metabolic disorders. As PCOS has much the same prevalence across the world, the excess economic burden attributable to PCOS globally is enormous, mandating that the scientific and policy community increase its focus on this important disorder.

**Funding:** The study was supported, in part, by PCOS Challenge: The National Polycystic Ovary Syndrome Association and by the Foundation for Research and Education Excellence

## Editor's evaluation

This important paper describes a valuable systematic review and meta-analysis of mental health problems in polycystic ovary syndrome (PCOS) that drive the excess economic burden associated with this common endocrine disorder. Interestingly, the cost of the diagnostic evaluation is only a relatively minor part of the total costs, but mental health disorders were identified as a significant component of the economic burden. These solid findings could not have been anticipated intuitively and are of considerable value for public health prioritization of PCOS.

## Introduction

Polycystic ovary syndrome (PCOS) is a highly prevalent disorder, highly inherited complex polygenic, multifactorial disorder (*Azziz et al., 2016*). PCOS is the single most common endocrine-metabolic disorder in reproductive-aged women today, affecting 5–15% of unselected reproductive-aged women (1990 National Institutes of Health criteria), and potentially represents a significant financial burden to our healthcare (*Azziz et al., 2005*; *Riestenberg et al., 2022*). Pathophysiological abnormalities in gonadotropin secretion or action, ovarian folliculogenesis, steroidogenesis, insulin secretion or action, and adipose tissue function, among others, have been described in PCOS. Women with PCOS are at increased risk for glucose intolerance and type 2 diabetes mellitus (T2DM), hepatic steatosis and metabolic syndrome, hypertension, dyslipidemia, vascular thrombosis, cerebrovascular accidents, possibly cardiovascular events, subfertility, and obstetric complications, endometrial carcinoma, and mood and psychosexual disorders (*Azziz et al., 2016*).

Although the physical symptoms of PCOS are increasingly recognized by practicing clinicians, little attention has focused on the psychological correlates of this frequent endocrine disorder (*Himelein and Thatcher, 2006*; *Brutocao et al., 2018*). A significant amount of research has been conducted showing the direct causal relationship between PCOS diagnosis and mental health (MH) disorders. PCOS is associated with an increased risk of a diagnosis of depression, anxiety, bipolar disorder, and obsessive–compulsive disorder (OCD) and is associated with worse symptoms of depression, anxiety, OCD, and somatization (*Himelein and Thatcher, 2006*; *Brutocao et al., 2018*). Screening for these disorders to allow early intervention may be warranted.

In order to allow for a more accurate prioritization of the disorder as a public health priority, we have pursued a comprehensive estimation of the economic burden of PCOS. In previous studies conducted by our team, we estimated the mean annual cost of the initial evaluation of PCOS to be $93 million, that of hormonally treating menstrual dysfunction/abnormal uterine bleeding to be $1.35 billion, that of providing infertility care to be $533 million, and that of treating hirsutism to be $622 million in 2014 USD (*Azziz et al., 2005*). In a more recent study, we estimated the mean annual costs of PCOS-associated T2DM and associated stroke to be $1.5 and $2.4 billion in 2020 USD, respectively (*Riestenberg et al., 2022*). The present study aims to assess the direct healthcare-related economic

**eLife digest** Polycystic Ovary Syndrome (PCOS) affects one in seven reproductive-age women worldwide. PCOS impacts women's physical and mental health, and it may also have detrimental effects on their social lives, academic achievement and careers. Studies show women with PCOS have higher rates of depression, anxiety, eating disorders, infertility and postpartum depression compared with women without the condition.

The economic burden of PCOS is enormous. Previous studies show PCOS-related economic costs totals billions of dollars. But few studies have examined the costs associated with PCOS-associated mental health care. Learning more about these costs may help policymakers and clinicians allocate resources for mental health care for women with PCOS.

Yadav et al. analyzed the results of 25 studies to assess the mental health impact of PCOS and its costs. The analysis found that women with PCOS are 60% more likely to have depression or anxiety compared to women without the condition. They were also twice as likely to have eating disorders. Caring for these mental health issues in PCOS patients increases US healthcare costs by approximately $4.2 billion yearly. These costs raise the healthcare-related economic burden of treating PCOS and associated conditions to $15 billion in the United States each year.

The analysis suggests that earlier recognition and better treatment of PCOS could reduce associated healthcare costs and improve the quality of life for women with PCOS. The results may help policymakers and clinicians understand the condition's impact and prioritize resources for PCOS care. More research on the condition is necessary to reduce the enormous economic and personal burden caused by it.

burden of PCOS-related MH disorders. To do so, we conducted a systematic review and meta-analysis of published studies with human subjects and controls that analyzed the relationship between MH disorders and previous diagnosis of PCOS, and then calculated the related direct economic burden.

## Materials and methods
### Systematic review
A systematic review was performed, adhering to the Preferred Reporting Items for Systematic Reviews and Meta-analyses (PRISMA) Statement and Checklist (https://www.prisma-statement.org/), for reports examining the relationship of MH disorders and PCOS through July 16, 2021. The systematic review was conducted on the six English databases (i.e., PubMed, Web of Science, EBSCO, Medline, Scopus, and PsycINFO). The following keywords were used (mental health OR mental illness OR mental disorder OR psych* OR anxiety OR depression OR quality of life OR eating disorder OR bulimia OR postpartum depression) AND ('cost* OR 'economic burden' OR 'cost-of-illness OR 'burden of illness'), "Depressive Disorder/economics"[MAJR], "PCOS" AND "economic burden" OR "costs" OR "cost-of-illness" OR "burden of illness", "PCOS" AND "economic burden" AND "mental health", 'polycystic ovary syndrome' AND 'anxiety, "Polycystic Ovary Syndrome/psychology"[MAJR] (*Supplementary file 1*; *Azziz, 2023*).

### Study eligibility criteria
Studies were eligible for inclusion if they: (1) were original peer-reviewed academic articles; and (2) were observational studies that presented accurate and precise data regarding the risk, including reporting relative risks, odds ratios (ORs), hazard ratios, or prevalence or incidence rates, of MH disorders in women with PCOS compared with a control group. MH disorders included depressive disorders, such as major depression disorder, dysthymia, minor depression or subclinical/subthreshold depression, or affective disorders containing depressive disorders, emotional distress, eating disorders, or mood and anxiety disorders. In the case of repeatedly published and studied literature based on the same batch of data or sample population, the most recent studies with the complete dataset were included. Studies were excluded if they: (1) were based on non-human species; (2) did not have full text available; (3) did not include a control group; (4) reported solely on diseases other than PCOS; or (5) were reviews, letters, or commentaries.

## Study selection and data extraction

Search strategy and study identification were performed by one investigator (S.Y.) using a standardized approach. Articles selected for inclusion were then screened by four authors (O.D., A.B., S.Y., and D.M.). Further, four investigators worked to independently extract data on study characteristics and outcomes (A.B., S.Y., O.D., and D.M.). Disagreements were discussed until consensus was reached.

## Quality assessment

Four investigators worked in duplicate to independently assess the quality of eligible studies using the Newcastle-Ottawa Scale (S.Y., O.D., A.B., and R.A.) (https://www.ohri.ca/programs/clinical_epidemiology/oxford.asp).

## Statistical analysis

Following the systematic search, the data were submitted to a meta-analysis to estimate the degree of relationship between PCOS and four dichotomous outcomes including anxiety, depression, eating disorders, and postpartum depression. The meta-analysis was performed using the DerSimonian–Laird random effects model (*DerSimonian and Laird, 1986*) and the results of the analyses were summarized using the pooled PR and its 95% confidence intervals (CIs) for each of the above outcomes.

We should note that when estimating risk, in general we can assume that the OR approximates the risk ratio when the prevalence in both groups is low (~<10%). However, when the prevalence of the outcome is higher the OR is likely to overestimate of risk ratio. Outcomes for this study were generally higher, so we used prevalence ratios (PRs) instead of ORs for the economic burden calculations. The limitation of using PRs is that these effects were not adjusted for the full set of covariates as for the OR analysis. However, most studies used groups that were matched by age and/or body mass index (BMI) by design.

Study-specific and overall effect estimates were visually presented using Forest plots. Between-study heterogeneity was evaluated using the $I^2$ statistic. In case of significant heterogeneity ($I^2$ >70%) sensitivity analyses were performed by excluding any outliers from the analysis. We defined an outlier as any study whose CIs did not overlap the CI of the pooled estimate for the purpose of the sensitivity analysis. To assess the potential impact of self-reporting on the outcomes observed, we also performed sensitivity analyses excluding studies where PCOS was self-reported. Publication bias was assessed using funnel plots. To adjust for possible publication bias, we recalculated the results using the 'trim and fill' method (*Shi and Lin, 2019*). In light of the asymmetry observed in the funnel plots, we further supplemented our assessment of potential publication bias by running Egger's test (*Egger et al., 1997*). Analyses were performed using R version 4.1.3.

When selecting measures of risk ratios, we noted that, in general, we can assume that the OR approximates the risk ratio when the prevalence in both groups is low (approximately <10%). However, when the prevalence of the outcome is higher than 10% the OR is likely to be an over-estimate of the risk ratio. As outcomes for this study were generally higher than 10%, we chose to use PRs instead of ORs for the economic burden calculations. These PRs were derived by comparing the prevalence in our study groups to the prevalence in the general population, which was determined based on control data in the included studies. The limitation of using PRs is that these effects were not adjusted for the full set of covariates as for the OR analysis. However, most studies used groups that were matched by age and/or BMI by design.

As previously (*Azziz et al., 2005*; *Riestenberg et al., 2022*), we defined the prevalence of PCOS based on the NIH criteria (phenotypes A and B of the Rotterdam criteria) or what is considered the 'classic' PCOS phenotypes, which we have conservatively estimated to be 6.6%. Annual cost data for medical treatment of depression for individuals 18–49 years of age was obtained from a 2021 study (*Greenberg et al., 2021*), for anxiety for ages 15–54 was obtained from a 1999 study (*Greenberg et al., 1999*), and for eating disorders for individuals 20–49 years of age was obtained from a 2021 study (*Streatfeild et al., 2021*).

The calculation of excess costs was based on cost estimates extracted from notable and frequently referenced studies. Though varying in publication year, these studies were selected based on their rigorous methodologies, comprehensive cost analyses, and alignment with our study's age-specific focus. For the medical treatment of depression in the age group 18–49, we relied on a recent study by *Greenberg et al., 2021*. This study thoroughly examines the economic burden associated with major depressive disorder

in adults, presenting up-to-date and in-depth cost analyses. The investigator's approach to segmenting the costs, which considers direct, indirect, and intangible costs, makes it an excellent choice for our study. The anxiety cost estimate for ages 15–54 was obtained from a seminal work by *Greenberg et al., 1999*. Despite being over two decades old, this paper remains a critical resource due to its exhaustive breakdown of anxiety-related care costs. This study is well regarded for its thoroughness and provides granular data on healthcare and medication costs and costs related to morbidity and mortality. The economic costs associated with eating disorders for ages 20–49 were sourced from the comprehensive study by *Streatfeild et al., 2021*, which examined the social and economic costs of eating disorders in the United States (U.S.), making it ideal for our research. We adjusted for inflation using the medical care inflation calculator (https://www.officialdata.org/Medical-care/price-inflation/).

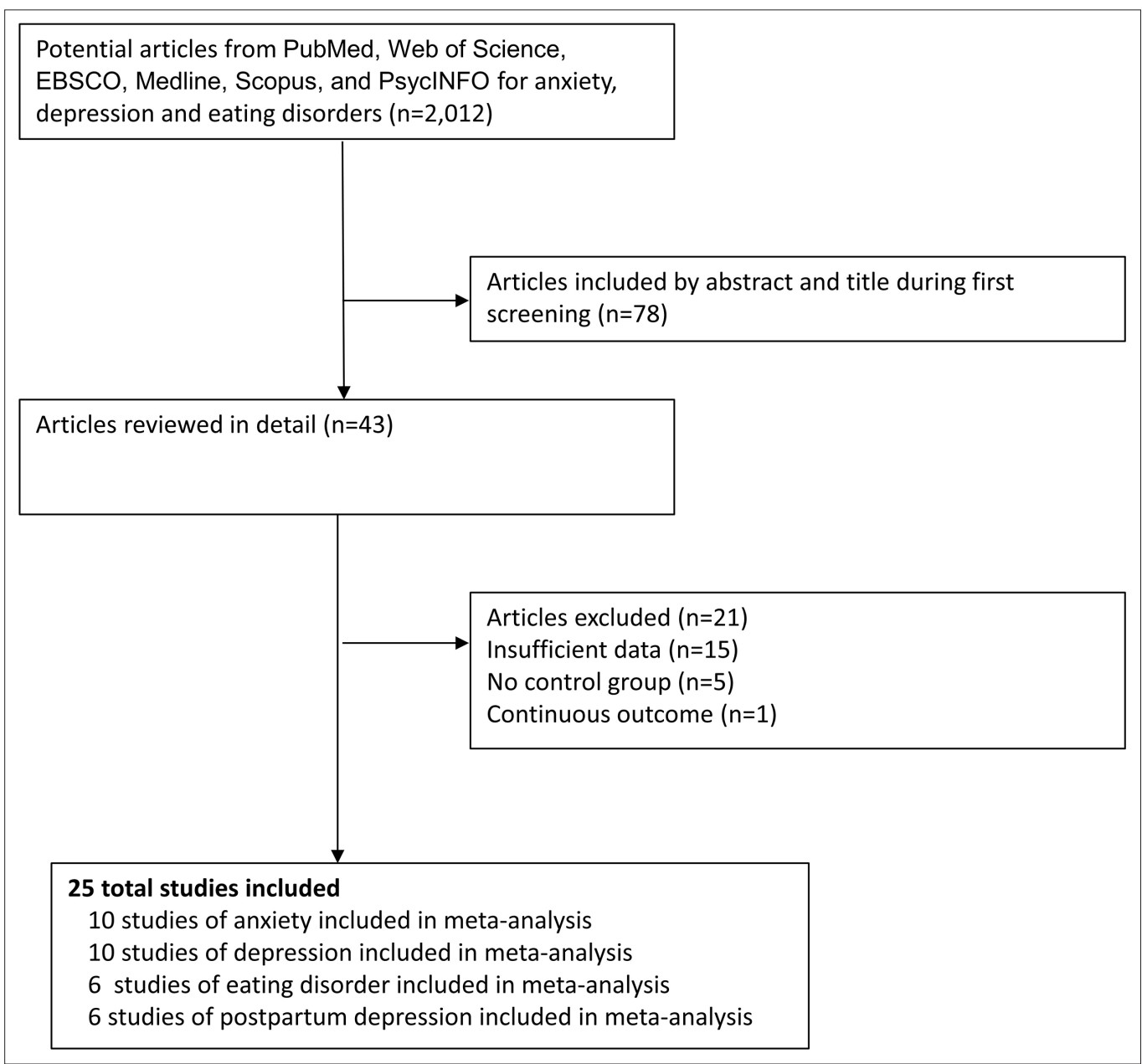

**Figure 1.** Flow diagram of the literature search and study selection process.

## Results

A total of 2018 studies were identified during the initial literature review, of which 78 were screened by title and abstract (*Figure 1*). Forty-three potentially eligible studies were reviewed in detail, of which 18 were excluded due to insufficient information. For example, some studies did not include information on a control group, did not include measures of association, used a continuous outcome instead of dichotomous outcome, or did not provide information about risk/prevalence that was needed to compute the PRs. The general characteristics of the 25 included studies (*Alur-Gupta et al., 2019*; *Tan et al., 2017*; *Karjula et al., 2017*; *Tay et al., 2020*; *Kaur et al., 2019*; *Asik et al., 2015*; *Cesta et al., 2016*; *Hussain et al., 2015*; *Månsson et al., 2008*; *Jedel et al., 2010*; *Li et al., 2017*; *Damone et al., 2019*; *Hollinrake et al., 2007*; *Pastore et al., 2011*; *Cinar et al., 2011*; *Adali et al., 2008*; *Lee et al., 2017*; *Pirotta et al., 2019*; *Koric et al., 2021*; *Joham et al., 2016*; *Muchanga et al., 2017*; *March et al., 2018*; *Tay et al., 2019*; *Alur-Gupta et al., 2021*; *Fugal et al., 2022*) are detailed in *Supplementary file 2a–d*.

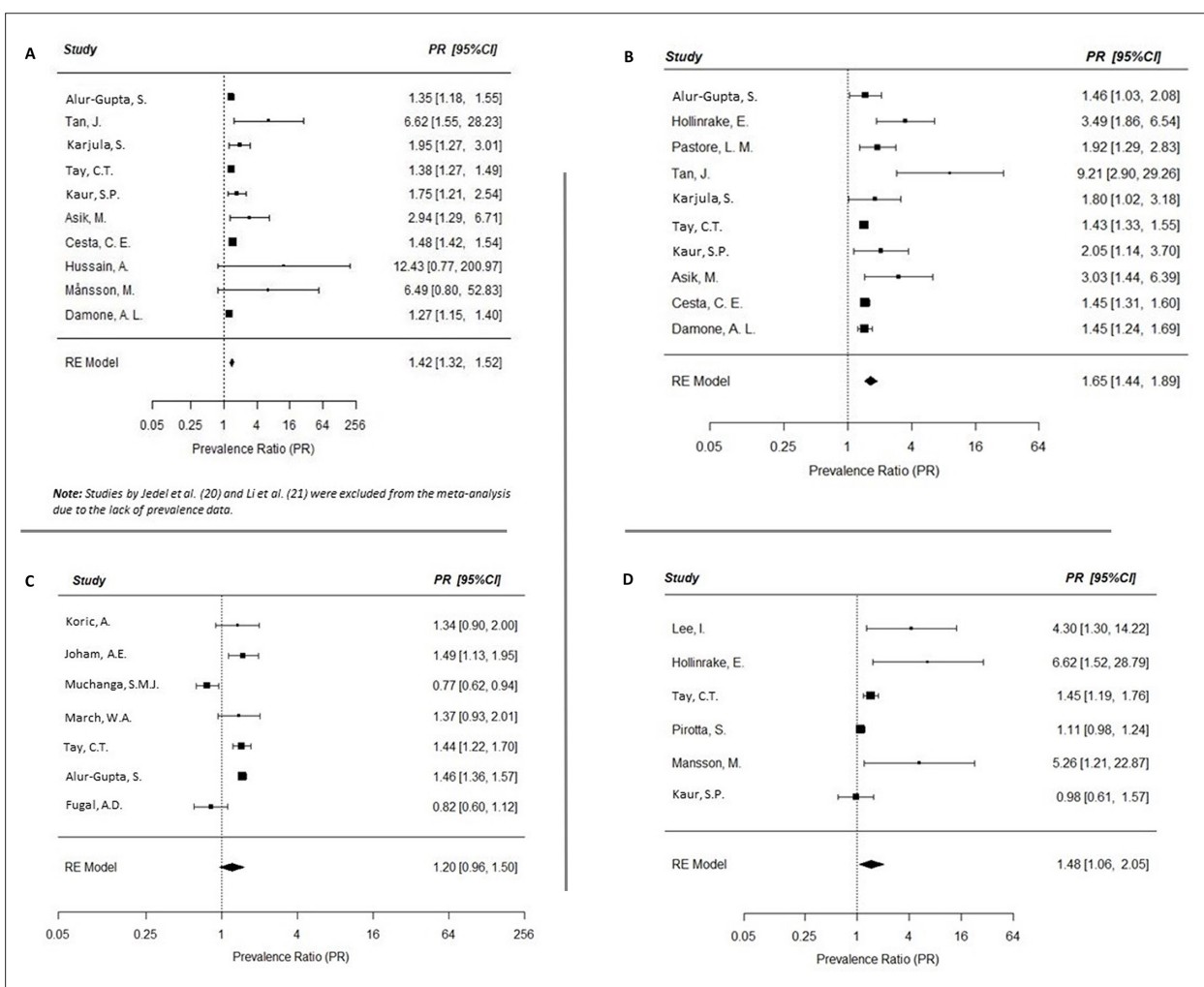

**Figure 2.** Meta-analyses of the prevalence of mental health disorders in women with polycystic ovary syndrome (PCOS). Forest plots (random effects model) of risk of mental health disorders in women with PCOS, including anxiety (**A**), depression (**B**), eating disorders (**C**), and postpartum depression (**D**). See text for abbreviations.

The online version of this article includes the following figure supplement(s) for figure 2:

**Figure supplement 1.** Trim and fill funnel plots for the meta-analysis of anxiety.

**Figure supplement 2.** Trim and fill funnel plots for the meta-analysis of depression.

**Figure supplement 3.** Trim and fill funnel plots for the meta-analysis of eating disorders.

## Excess prevalence of anxiety in women with PCOS

Twelve studies, all assessed as 'high quality' (*Supplementary file 2e*), were initially identified, measuring an association between anxiety and PCOS (*Alur-Gupta et al., 2019*; *Tan et al., 2017*; *Karjula et al., 2017*; *Tay et al., 2020*; *Kaur et al., 2019*; *Asik et al., 2015*; *Cesta et al., 2016*; *Hussain et al., 2015*; *Månsson et al., 2008*; *Jedel et al., 2010*; *Li et al., 2017*; *Damone et al., 2019*). However, only ten studies were included in the meta-analysis, as two studies were excluded due to the lack of prevalence data (*Jedel et al., 2010*; *Li et al., 2017*). Compared to age-matched women without PCOS, those with PCOS had a higher prevalence of anxiety in the meta-analysis (random effects PR: 1.42; 95% CI: 1.32, 1.52; $I^2$ 43.82%; *Figure 2A*). For our economic burden calculations, we considered PCOS patients as having a 1.42-fold greater risk of anxiety compared to those without PCOS.

Considering that comparably aged women of the general population have a prevalence of anxiety of 9.15%, the overall prevalence of anxiety in women with PCOS can be estimated to be 1.42 × 9.15% = 12.99%. The excess prevalence of anxiety due to PCOS is therefore 12.99–9.15% = 3.84%; the excess number of anxiety cases due to PCOS is 5,631,459 × 3.84% = 216,248 individuals.

## Excess prevalence of depression in women with PCOS

While 15 studies were initially identified (*Alur-Gupta et al., 2019*; *Tan et al., 2017*; *Karjula et al., 2017*; *Tay et al., 2020*; *Kaur et al., 2019*; *Asik et al., 2015*; *Cesta et al., 2016*; *Hussain et al., 2015*; *Månsson et al., 2008*; *Damone et al., 2019*; *Hollinrake et al., 2007*; *Pastore et al., 2011*; *Cinar et al., 2011*; *Adali et al., 2008*), we used only the 10 studies assessed as being of 'high quality' (*Supplementary file 2e*) for the meta-analysis. Compared to age-matched women without PCOS, those with PCOS had a higher prevalence of depression in the meta-analysis (random effects PR: 1.65; 95% CI: 1.44, 1.89; $I^2$ 63.0%; *Figure 2B*). For our economic burden calculations, we considered PCOS patients as having a 1.65-fold greater risk of depression compared to those without PCOS.

Considering that comparably aged women of the general population have a prevalence of depression of 8.9%, the overall prevalence of depression in women with PCOS can be estimated to be 1.65 × 8.9% = 14.69%. The excess prevalence of depression due to PCOS is therefore 14.69–8.9% = 5.79%; the excess number of depression cases due to PCOS is 4,528,088 × 5.79% = 262,176 individuals.

## Excess prevalence of eating disorders in women with PCOS

Six studies, all assessed as 'high quality' (*Supplementary file 2e*), were included for an association between eating disorders and PCOS (*Karjula et al., 2017*; *Tay et al., 2020*; *Hussain et al., 2015*; *Damone et al., 2019*; *Adali et al., 2008*; *Lee et al., 2017*). Compared to age-matched women without PCOS, those with PCOS had a higher prevalence of eating disorders in the meta-analysis (random effects PR: 1.48; 95% CI: 1.06, 2.05; $I^2$ 74.41%; *Figure 2C*).

Considering that comparably aged women of the general population have a prevalence of eating disorders of 2.4%, the overall prevalence of eating disorders in women with PCOS can be estimated to be 1.48 × 2.4% = 3.55%. The excess prevalence of depression due to PCOS is therefore 3.55–2.4% = 1.15%. Therefore, the excess number of anxiety due to PCOS is 4,240,306 × 1.15% = 48,764 individuals.

**Table 1.** Estimates of the excess prevalence and economic burden associated with mental health (MH) morbidities of polycystic ovary syndrome (PCOS) as of 2021 in the United States.

| MH morbidities | Excess prevalence of morbidity in PCOS (%) | Annual costs in billions in 2021 USD (% of total costs in category) |
|---|---|---|
| Anxiety | 3.84 | $1.939 (45.5) |
| Depression | 5.79 | $1.678 (39.4) |
| Eating disorders | 1.15 | $0.644 (15.1) |
| Total excess cost of MH disorders in PCOS | | 4.261 (100) |

## Excess prevalence of postpartum depression in women with PCOS

Six studies, all of high quality (*Supplementary file 2e*), were included examining the association between postpartum depression and PCOS (*Koric et al., 2021*; *Joham et al., 2016*; *Muchanga et al., 2017*; *March et al., 2018*; *Tay et al., 2019*; *Alur-Gupta et al., 2021*; *Fugal et al., 2022*). Compared to age-matched women without PCOS, those with PCOS had a higher prevalence of postpartum depression in the meta-analysis, however this association did not reach statistical significance (random effects PR: 1.20; 95% CI: 0.96, 1.50; *Figure 2D*). Because the association between PCOS and postpartum depression was not found to be statistically significant in this meta-analysis, postpartum depression was excluded from our calculations of economic burden.

## Economic burden of anxiety in women with PCOS

The cost of anxiety-related care per individual in need was estimated to be $2694 in 1990, which converts to $8966 in 2021 USD. Therefore, the excess cost of anxiety-related care in PCOS is 216,248 × $8966 = $1,938,879,568 USD in 2021 (*Table 1*).

## Economic burden of depression in women with PCOS

The cost of depression-related care per individual in need was estimated to be $5726 in 2018, which converts to $6401 in 2021 USD. Therefore, the excess cost of depression-related care in PCOS is 262,176 × $6401 = $1,678,188,576 USD in 2021 (*Table 1*).

## Economic burden of eating disorders in women with PCOS

The cost of eating disorder-related care per individual in need was estimated to be $11,808 in 2018, which converts to $13,200 in 2021 USD. Therefore, the excess cost of eating disorder-related care in PCOS is 48,764 × $13,200 = $643,684,800 USD in 2021 (*Table 1*).

## Assessing for the impact of self-reporting in PCOS

We repeated the analyses after excluding studies where PCOS was self-reported (*Supplementary file 3a, b*). We observed that the point estimate for anxiety remained consistent with our original findings. Conversely, the point estimates for depression and eating disorders were slightly increased. Nevertheless, compared to our primary analysis, these variations were within the range of random variation, underscoring the robustness of our initial findings.

## Assessing for potential publication bias

To assess for possible publication bias, we recalculated the results using the 'trim and fill' method. In the 'trim and fill' analysis for PCOS-related anxiety the estimated number of missing studies was 4 and the corresponding pooled random effects PR estimate was 1.40 (95% CI: 1.31, 1.50, p < 0.001) (*Figure 2—figure supplement 1*). In the 'trim and fill' analysis for PCOS-related depression the estimated number of missing studies was 4 and the corresponding pooled random effects PR estimate was 1.50 (95% CI: 1.28, 1.76, p < 0.0001) (*Figure 2—figure supplement 2*). In the 'trim and fill' analysis for PCOS-related eating disorders the estimated number of missing studies was 2 and the corresponding pooled random effects PR estimate was 1.30 (95% CI: 0.92, 1.85; p = 0.1371) (*Figure 2—figure supplement 3*). We did not analyze the data for PCOS-related postpartum depression using the 'trim and fill' approach as these results were already not significant.

That the results for PCOS-related eating disorders are no longer significant after applying the 'trim and fill' adjustment means that these results were sensitive to one type of selection bias that is due 'small study' effects, that is, the tendency of small studies to suppress publication of results that are negative. However, we should note that the 'trim and fill' method cannot be used as formal proof for the presence of publication bias due to 'small study' effects, as it is possible that there are other explanations for the lack of symmetry on the funnel plots, including heterogeneity of study populations, covariates, or outcome definitions that may give rise to a lack of symmetry. However, based on this analysis it appears that the results for this outcome are not as robust as for the other outcomes.

The 'trim and fill' method suggested that the results for anxiety and depression were robust to this type of publication bias. In contrast, results for eating disorders were somewhat less robust. In order to assess publication bias, we also conducted the Egger's test for funnel plot asymmetry. The results were significant for depression, anxiety, and eating disorders, implying potential publication

**Table 2.** Direct healthcare-related economic burden in polycystic ovary syndrome (PCOS) as of 2021 in the United States.

| Process/disorder | Original year economic burden published (reference) | Economic burden year of publication (in billions)* | Economic burden in 2021 USD (in billions) | % of total economic burden |
|---|---|---|---|---|
| Initial evaluation | 2004 (2) | $0.093 | 0.166 | 1.09 |
| Menstrual dysfunction/AUB | 2004 (2) | $1.350 | 2.408 | 15.88 |
| Infertility care | 2004 (2) | $0.533 | 0.951 | 6.27 |
| Hirsutism | 2004 (2) | $0.622 | 1.109 | 7.31 |
| GDM† | 2020 (3) | $0.672† | 0.684 | 4.51 |
| gHTN† | 2021 (3) | $0.208† | 0.212 | 1.40 |
| Preeclampsia† | 2022 (3) | $0.137† | 1.400 | 9.23 |
| T2DM | 2023 (3) | $1.500 | 1.527 | 10.07 |
| Stroke | 2024 (3) | $2.400 | 2.445 | 16.12 |
| Anxiety | 2021 | Present study | 1.939 | 12.79 |
| Depression | 2021 | Present study | 1.678 | 11.07 |
| Eating disorders | 2021 | Present study | 0.644 | 4.25 |
| **Total** | | | **15.163** | 100.00 |

*Updated for inflation using medical CPI (https://www.in2013dollars.com/Medical-care-services/price-inflation).
†Estimates of economic burden in prior publication updated for inflation using medical CPI (https://www.in2013dollars.com/Medical-care-services/price-inflation).

bias arising from 'small study' effects (***Supplementary file 3c***). Nonetheless, alternative explanations of funnel plot asymmetry are possible. For example, funnel plot asymmetry can be caused by between-study heterogeneity. Different studies may estimate different effects due to differences in study design, characteristics of the study sample, outcome definitions, or geographic location. For depression, the removal of one outlier study by ***Tan et al., 2017*** from the meta-analysis eliminated about 30% of the between-study heterogeneity.

## Discussion

In the U.S., the additional direct healthcare costs associated with MH disorders in PCOS were estimated to be $1.939 billion/yr., $1.678 billion/yr., and $0.644 in 2021 USD for anxiety, depression, and eating disorders, respectively. The combined additional direct healthcare costs associated with depression and anxiety in PCOS was estimated to be $4.261 billion/yr. in 2021 USD, of which 45% can be attributable to anxiety, 40% to depression, and the remainder to eating disorders. While the prevalence of postpartum depression appeared to be increased in PCOS, the difference did not reach significance on meta-analysis and this outcome was not included in our economic burden calculations.

Taken together, including our prior economic burden assessments (***Azziz et al., 2005***; ***Riestenberg et al., 2022***), the total excess economic burden estimated for PCOS exceeds $15 billion annually in 2021 USD (***Table 2***). Of this cost, approximately 28% will be accounted for the cost of treating PCOS-related MH disorders, including anxiety, depression, and eating disorders; 29.5% will be accounted for the cost of treating reproductive endocrine morbidities (menstrual dysfunction/abnormal uterine bleeding, hirsutism, and infertility); 15.1% is attributable to obstetrical and pregnancy related disorders; and 10.1% and 16.1% is attributable to T2DM and strokes, respectively. The cost of the initial diagnostic evaluation of PCOS is very low ($166 million annually in 2021 USD; ***Table 2***), accounting for only 1.1% of the total economic burden attributable to direct healthcare costs of the disorder estimated so far. These data strongly suggest that ensuring quality diagnosis and evaluation for all patients with PCOS is a cost-effective approach to ameliorating the complications and costs associated with the disorder.

For perspective, the estimated direct healthcare costs attributable to ovarian cancer, lung cancer, prostate cancer, and breast cancer, in the U.S. are $7.8, $16.8, $26.7, and $20.5 billion in 2021 USD, respectively (*Waters and Graf, 2018*), compared to $15.2 billion for PCOS so far. Furthermore, the included costs are only for those morbidities that to date have been confirmed as increased in PCOS relative to controls after careful meta-analyses considering the quality of the studies. As further studies are undertaken it is likely that the economic burden of PCOS related to direct healthcare costs will continue to rise.

Liu et al. assessed the current burden of PCOS at the global, regional, and country-specific levels in 194 countries and territories according to age and sociodemographic index (SDI) (*Liu et al., 2021*). The investigators used data from the Global Burden of Diseases, Injuries and Risk Factors Study (GBD) 2017 to estimate the total and age-standard PCOS incidence rates and the associated disability-adjusted life-years (DALYs) rates among women of reproductive age in both 2007 and 2017, and the trends in these parameters from 2007 to 2017. The data sources used in GBD take many forms, including census data, vital registrations, disease registries, survey data, and published and unpublished scientific literature, among other sources (https://www.healthdata.org/acting-data/what-data-sources-go-gbd). These investigators concluded that PCOS accounted for 0.43 million associated DALYs. They also noted slight increases in the age-standardized incidence of PCOS and DALYs among women of reproductive age (15–49 years) from 2007 to 2017 at the global level, and in most regions and countries. Safiri et al. also used the GDB Study 2017 database to determine the global, regional, and national burden of PCOS, by age and SDI, over the period 1990–2019 (*Safiri et al., 2022*). These investigators reported that in 2019 the global age-standardized point prevalence and annual incidence rates for PCOS were 1677 (1.7%) and 59 (0.06%) per 100,000, respectively.

Neither one of these studies estimated the economic cost of the burden observed. Furthermore, we should note that (*Liu et al., 2021*) reported only on global age-standardized PCOS incidence rates (i.e., the occurrence of *new cases* of PCOS over a specified period of time), not prevalence rates (i.e., the total proportion of persons in the population who have PCOS at a specified point in time or over a specified period of time), among women of reproductive age. While *Safiri et al., 2022* present an estimate of prevalence, we should note that the estimate is significantly lower than that reported when populations are assessed directly for the prevalence of PCOS (1.7% vs. 6.6% or greater), likely reflecting the chronic underdiagnosis of PCOS.

Ding et al. estimated the burden of disease attributable to type 2 DM in women with PCOS using individual patient data from a UK primary care database between 2004 and 2014 and aggregate data from the literature to obtain conversion rates through disease progression (*Ding et al., 2018*). A simulation approach was applied to model the population dynamics of PCOS over a follow-up period of 25 years in using Bayesian modeling. The investigators estimated that the associated annual healthcare burden of T2DM in PCOS was at least £237 million in 2014 pounds in the UK. Taking into account the relative populations of reproductive-aged women (15–49 years) of the UK and the U.S. (14.7 vs. 76.5 million) (*United Nations, Department of Economic and Social Affairs, Population Division, 2022*), the healthcare inflation rate since 2014 (19.2%), and the conversion rate of pounds to dollars in mid-2021 (0.72 USD to 1 pound), the economic burden for PCOS-associated T2DM estimated by *Ding et al., 2018* is equivalent to $2.04 billion 2021 USD, somewhat higher than the $1.527 2021 USD that we previously estimated (*Table 2*; *Riestenberg et al., 2022*).

The strength of this study lies in the exclusive inclusion of peer-reviewed and controlled studies. However, our analysis encountered limitations due to the restricted number of studies that satisfied the inclusion criteria. This could be due to the fact that PCOS diagnosis creates an umbrella effect that encompasses physical as well as MH disease which creates a false belief in patients that they cannot be treated or diagnosed with a disease other than PCOS due to the misconception that their symptoms stem solely from the PCOS diagnosis. Other limitations of this study include the fact that while the cost estimates were conducted for the U.S., the meta-analysis used some studies from other countries.

Additional possible limitations stem from the inclusion of studies that relied on self-reported PCOS, which could introduce recall and reporting bias, potentially affecting the reported PRs of MH-related conditions. However, we should note that the point estimates changed little when excluding studies that used self-reported PCOS. In our calculations, we used inflation-adjusted estimates from previous studies to account for the changing value of currency over time. While this method provides a useful

approximation, we acknowledge that it does not account for changes in treatment practices, health-care policies, or the overall structure of healthcare costs that might have occurred since the original estimates were made. Finally, while other MH disorders have been shown to be associated with PCOS, such as bipolar disorder and OCD, we focused our study solely on the four most common disorders found in the literature (*Himelein and Thatcher, 2006*; *Brutocao et al., 2018*).

In conclusion, our current study suggests that the additional direct healthcare costs due to PCOS-related anxiety, depression, and eating disorders exceeds $4 billion annually in 2021 USD in the U.S. So far, the total direct economic burden of PCOS exceeds $15 billion annually in 2021 USD, and MH disorders account for almost one-third of these costs. Notably, these estimates are solely for the U.S. and PCOS is a global disease. As PCOS is clinically most apparent during the reproductive age, we should note that the number of women between the ages of 15 and 49 worldwide is estimated to be about 1.9 billion (*United Nations, Department of Economic and Social Affairs, Population Division, 2022*). Taking into account that most studies assessing the prevalence of PCOS report a minimum rate similar to what was used on the present study (i.e., 6.6% based on the NIH criteria) (*Bozdag et al., 2016*), we can estimate that there are *at least* 125.4 million women affected with PCOS worldwide. Consequently, the worldwide economic burden for PCOS can be assumed to be enormous.

While we have examined the direct economic burden related to the medical or health-related costs of the disorder, we should note that a complete understanding of the economic burden of the disorder requires that we also assess the indirect (those attributable to loss of work productivity) and intangible (those related to the pain and sufferings of patients because of a disorder, usually measured by using the reduction in quality of life) costs of the disorder. Our data suggest that improved detection of PCOS and more significant clinical awareness and interventions for PCOS and its associated disorders is a cost-effective approach to ameliorating the economic, health, and quality of life impact of PCOS. Finally, our findings further highlight the need to increase investment in PCOS research, which is currently severely underfunded relative to the economic burden of the disorder (*Brakta et al., 2017*).

# Additional information

## Competing interests

William Patterson: is a Director of Public Affairs, Study Recruitment and Patient Registry Manager, and Intern Preceptor for PCOS Challenge: The National Polycystic Ovary Syndrome Association. The author has no other competing interests to declare. Sasha Ottey: received honoraria from Institute for Family Centeredness; has received travel support for attendance at Women's Health Innovation Summit; and is Executive Director for PCOS Challenge: The National Polycystic Ovary Executive Director Syndrome Association. The author has no other competing interests to declare. Ricardo Azziz: received a grant from Ferring Pharmaceuticals; has received royalties from Wolters Kluwer Health, Johns Hopkins University Press, Springer, and McGraw Hill; has received honoraria from Davidson-Mestman course; has participated on a data safety monitoring board for University of Michigan and Guangzhou Medical University; has held a leadership or fiduciary role at American Society for Reproductive Medicine; holds stock at Martin Imaging and Arora Forge; has received consulting fees from Spruce Biosciences, Fortress Biotech, Rani Therapeutics, Core Access Surgical for PCOS, female reproduction and gynecologic surgery; and serves as Senior Editor at eLife. The author has no other competing interests to declare. The other authors declare that no competing interests exist.

## Funding

| Funder | Grant reference number | Author |
|---|---|---|
| Foundation for Research and Education Excellence | | Ricardo Azziz |
| PCOS Challenge: The National Polycystic Ovary Syndrome Association | | Surabhi Yadav |

| Funder | Grant reference number | Author |
| --- | --- | --- |

The funders had no role in study design, data collection, and interpretation, or the decision to submit the work for publication.

## Author contributions

Surabhi Yadav, Conceptualization, Data curation, Software, Formal analysis, Validation, Writing - original draft; Olivia Delau, Data curation, Software, Formal analysis, Validation, Investigation, Methodology, Writing - original draft; Adam J Bonner, Data curation, Formal analysis, Validation, Investigation, Writing – review and editing; Daniela Markovic, Software, Formal analysis, Validation, Methodology, Writing – review and editing; William Patterson, Conceptualization, Resources, Methodology, Project administration, Writing – review and editing; Sasha Ottey, Conceptualization, Resources, Supervision, Investigation, Writing – review and editing; Richard P Buyalos, Conceptualization, Resources, Funding acquisition, Project administration, Writing – review and editing; Ricardo Azziz, Conceptualization, Resources, Data curation, Formal analysis, Supervision, Validation, Investigation, Methodology, Project administration, Writing – review and editing

## Author ORCIDs

Ricardo Azziz ⬤ https://orcid.org/0000-0002-3917-0483

## Ethics

This study was exempt from ethics approval as the investigators collected and synthesized data from previous published studies in which informed consent has already been obtained by the trial investigators.

## Decision letter and Author response

Decision letter https://doi.org/10.7554/eLife.85338.sa1
Author response https://doi.org/10.7554/eLife.85338.sa2

# Additional files

## Supplementary files
• MDAR checklist

• Supplementary file 1. Search terms used for systematic review.

• Supplementary file 2. Characteristics and quality of studies included. (a) Characteristics of included studies categorized by anxiety; (b) characteristics of included studies categorized by depression; (c) characteristics of included studies categorized by eating disorders; (d) characteristics of included studies categorized by postpartum depression; and (e) quality scores of studies using Newcastle-Ottawa Scale.

• Supplementary file 3. Sensitivity analysis and Egger's test for studies included. (a) Sensitivity analysis: all studies including self-reported polycystic ovary syndrome (PCOS); (b) sensitivity analysis: excluding studies with self-reported PCOS; and (c) Egger's test results for assessing funnel plot asymmetry.

## Data availability

The data is available at Zenodo: https://doi.org/10.5281/zenodo.8122261.

The following dataset was generated:

| Author(s) | Year | Dataset title | Dataset URL | Database and Identifier |
| --- | --- | --- | --- | --- |
| Azziz R | 2023 | Direct Economic Burden of Mental Health Disorders Associated with Polycystic Ovary Syndrome: Systematic Review and Meta-analysis (Rev. 3) | https://doi.org/10.5281/zenodo.8122261 | Zenodo, 10.5281/zenodo.8122261 |

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
