## [Editor Report]

This important paper describes a valuable systematic review and meta-analysis of mental health problems in polycystic ovary syndrome (PCOS) that drive the excess economic burden associated with this common endocrine disorder. Interestingly, the cost of the diagnostic evaluation is only a relatively minor part of the total costs, but mental health disorders were identified as a significant component of the economic burden. These solid findings could not have been anticipated intuitively and are of considerable value for public health prioritization of PCOS.

---

## [Decision Letter]

**Decision letter after peer review:**

Thank you for submitting your article "Direct Economic Burden of Mental Health Disorders Associated with Polycystic Ovary Syndrome: Systematic Review and Meta-analysis" for consideration by *eLife*. Your article has been reviewed by two peer reviewers, and I have overseen the evaluation in my dual role of Reviewing Editor and Senior Editor. The following individuals involved in the review of your submission have agreed to reveal their identities: Michel Pugeat (Reviewer #1); Rebecca Elaine Campbell (Reviewer #2).

Essential revisions:

*Reviewer #2 (Recommendations for the authors):*

– P3, ln 59 'Following' vs Followed.

– P5, ln 98 'PCOS is the single…' vs PCOS single.

– P6, ln 124 add 'mean annual' to costs of PCOS-associated…

– P10, ln 215 'an' vs as.

– P16, ln 381 'mental health disease' vs mental disease.

– P16, ln 379- consider revising this sentence for clarity.

– P17, ln 17 remove 'with.'

– Given the asymmetric nature of the funnel plots, an Egger's test could also be run for additional assessment of publication bias in addition to the trim and fill.

---

## [Author Response]

Essential revisions:Reviewer #2 (Recommendations for the authors):– P3, ln 59 'Following' vs Followed.– P5, ln 98 'PCOS is the single…' vs PCOS single.– P6, ln 124 add 'mean annual' to costs of PCOS-associated…– P10, ln 215 'an' vs as.– P16, ln 381 'mental health disease' vs mental disease.– P16, ln 379- consider revising this sentence for clarity.– P17, ln 17 remove 'with.'– Given the asymmetric nature of the funnel plots, an Egger's test could also be run for additional assessment of publication bias in addition to the trim and fill.

We conducted Egger’s test for funnel plot asymmetry. The results were significant for depression, anxiety, and eating disorders, implying potential publication bias arising from "small study" effects. Nonetheless, other factors, such as heterogeneity of study populations, covariates, or outcome definitions, could also account for the funnel plot asymmetry. The "trim and fill" method suggested that the results for anxiety and depression were robust to this type of publication bias. In contrast, results for eating disorders were somewhat less robust (Line 335; Supplemental Table 9). Please note that the power of this method to detect bias will be low with small numbers of studies. We have added this data to the Results and Discussion sections of our manuscript.